# Quantum physics in connected worlds

Joseph Tindall [1,2] ✉, Amy Searle[2], Abdulla Alhajri[2,3] & Dieter Jaksch [2,4,5]

Theoretical research into many-body quantum systems has mostly focused on regular structures which have a small, simple unit cell and where a vanishingly small fraction of the pairs of the constituents directly interact. Motivated by advances in control over the pairwise interactions in many-body simulators, we determine the fate of spin systems on more general, arbitrary graphs. Placing the minimum possible constraints on the underlying graph, we prove how, with certainty in the thermodynamic limit, such systems behave like a single collective spin. We thus understand the emergence of complex many-body physics as dependent on 'exceptional', geometrically constrained structures such as the low-dimensional, regular ones found in nature. Within the space of dense graphs we identify hitherto unknown exceptions via their inhomogeneity and observe how complexity is heralded in these systems by entanglement and highly non-uniform correlation functions. Our work paves the way for the discovery and exploitation of a whole class of geometries which can host uniquely complex phases of matter.

Research into many-body quantum physics has predominantly involved setups with local interactions and a high degree of spatial symmetry. Such a focus is natural, with the short-range, homogeneous nature of the resulting Hamiltonian being a reasonable reflection of reality in naturally occurring materials, and also beneficial, since such features can be exploited in order to render the Hamiltonian soluble with computational methods.

Recent experimental advances, however, have made it clear that many-body quantum physics need not be limited to such geometries. In Rydberg simulators[1–5], for example, free placement of the individual atoms is now possible using optical tweezers. Moreover, in a range of other platforms—which include atoms trapped in cavities[6] or photonic waveguides[7], Moiré Heterostructures[8], trapped ions[9] and superconducting circuits[10]—experimentalists are demonstrating increasing control over the pairwise interactions and geometries in the Hamiltonians that they can realise. For instance, proposals now exist to use trapped ion arrays to engineer many-body spin Hamiltonians defined over arbitrary graphs[11,12] whilst a recent experiment successfully probed the out-of-equilibrium behaviour of a spin model with all-to-all interactions[13].

In general, the limitations on the geometries which can be realised are continually being lowered and open up the tantalising possibility of

exploring and utilising many-body quantum physics on a wide range of complex graph structures, including those which are well-established in the social[14] and biological[15,16] sciences.

Despite this experimental progress, from a theoretical perspective, there is little understanding of the physics of many-body Hamiltonians when hosted on structures that are not either all-to-all setups[17] or sparsely connected, low-dimensional lattices. The last decade has seen significant interest in low-dimensional lattices with long-range interactions[18–22], yet despite the increased connectivity, the underlying system is still translationally invariant. The fate of many-body physics on more general structures is unknown.

In this work, we rectify this by approaching the many-body problem in an entirely new way. We take a generic spin $s$ Hamiltonian which encompasses a wide range of celebrated many-body models and treat the geometry as a parameter itself, encoding it in an underlying graph upon which the spins reside and interact via the edges. We then uncover the physics of the system when placing various levels of constraints on the graph. First, we place the minimal possible constraints on the graph and prove that in thermal equilibrium for a graph chosen uniformly at random from all possible simple graphs, almost surely, there is an absence of many-body physics, and only collective, mean-field physics is possible. We achieve this by proving that, with

[1]Center for Computational Quantum Physics, Flatiron Institute, 162 5th Avenue, New York, NY 10010, USA. [2]Clarendon Laboratory, University of Oxford, Parks Road, Oxford OX1 3PU, UK. [3]Technology Innovation Institute, Masdar City 9639 Abu Dhabi, United Arab Emirates. [4]The Hamburg Centre for Ultrafast Imaging, Universität Hamburg, Luruper Chaussee 149, 22761 Hamburg, Germany. [5]Institut für Laserphysik, Universität Hamburg, Luruper Chaussee 149, 22761 Hamburg, Germany. ✉e-mail: jtindall@flatironinstitute.org

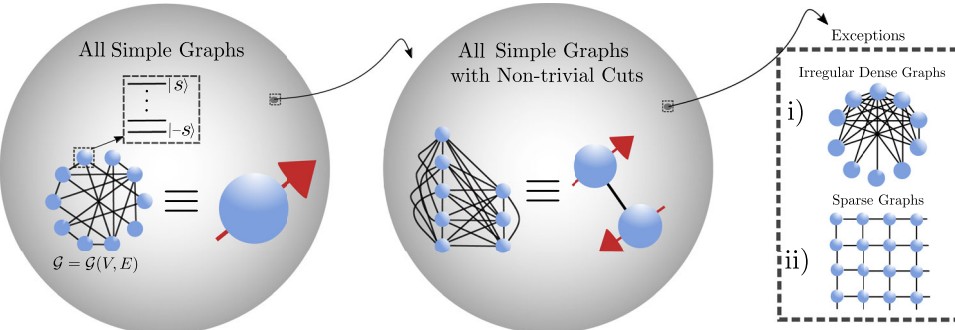

**Fig. 1 | In this work we consider a many-body spin $s$ Hamiltonian—see Eq. (1)—defined over a simple graph $\mathcal{G}(V,E)$ where the vertices represent the spins and the edges the pairs of spins upon which the two-body terms act.** We prove for a graph chosen uniformly at random from all simple graphs that, as the graph size $L$ increases, the equilibrium properties of the system become increasingly like that of a single collective spin and any many-body effects vanish as $L \to \infty$. In order for this not to be true, the graph must possess a non-trivial cut and we prove that even for such a graph, chosen at random amongst all those with a non-trivial cut, the system can effectively be reduced to that of two interacting collective spins. The emergence of complex, non-collective physics is thus dependent on more structured, 'exceptional' graphs which exist in a vanishingly small subspace of the space of all simple graphs. These include the well-known sparse, regular structures that arise in nature and a new class of graphs we identify here: irregular dense structures.

increasing certainty and accuracy as the graph size $L$ increases, the free energy density on such a graph can be approximated with that for a single collective spin, as indicated in Fig. 1.

Our result is based on the fact that, for the Erdős–Rényi (ER) graph as $L \to \infty$, there exists no partition (cut) where the number of edges between the partitions differs significantly from its expected value. We then show that even for random graphs constrained to have a non-trivial cut the system can asymptotically almost surely (i.e. with probability tending to 1 as $L \to \infty$) be reduced to a pair of interacting, large spins—a result also depicted in Fig. 1. The emergence of complex, many-body physics is thus dependent on exceptional, highly structured graphs which are strongly distinct from almost all other graphs. We discuss how the sparse, regular lattices which commonly arise in nature are such an exception and go on to discover a hitherto unknown class of exceptional graphs that violate our proofs and where complex, many-body physics emerges: irregular dense graphs.

We illustrate these results. Taking the limit of our Hamiltonian which involves the competition between anti-ferromagnetic and $XY$ couplings, we demonstrate how the nature of the underlying phase transition changes from non-existent, to first-order, to second-order, when considering random graphs chosen from the space of all graphs, all graphs with a non-trivial cut and dense inhomogeneous graphs, respectively. Using a well-established measure from image classification we demonstrate how, for the irregular dense graphs, the second-order phase transition coincides with significant complexity in images of the off-diagonal correlations in the system. Such a feature, which is not specific to the limit taken on our Hamiltonian, highlights the uniqueness of the ground state on these structures and cannot occur in sparse, translationally invariant structures. Our work here establishes the fate of many-body physics on a wide range of graphs and uncovers a new class of structures where a novel, non-collective states of matter can emerge.

## Results
### Model and Hamiltonian
We start by defining an arbitrary simple graph via $\mathcal{G} = \mathcal{G}(V, E)$ where $V$ is the $L = |V|$ vertices and $E$ is the $N_E = |E|$ edges. On each vertex of the graph, we place a spin $s$ particle and have these spins interact with each other via the unweighted edges of the graph and be affected by a global field. The Hamiltonian for the total energy reads

$$\hat{H}(\mathcal{G}) = \frac{L}{N_E}\left(\sum_{(v,v')\in E} \hat{h}_{v,v'}\right) + \sum_{v \in V} \hat{h}_v, \tag{1}$$

with $\hat{h}_{v,v'}$ and $\hat{h}_v$ being, respectively, one and two-body operators acting on the subscripted vertices. We take $\hat{h}_{v,v'} = J_x \hat{s}_v^x \hat{s}_{v'}^x + J_y \hat{s}_v^y \hat{s}_{v'}^y + J_z \hat{s}_v^z \hat{s}_{v'}^z$ where $J_x, J_y, J_z \in \mathbb{R}$. The canonical spin operator $\hat{s}_v^\alpha$ acts on spin $V$ in the $\alpha = x, y$ or $z$ direction. We also set $\hat{h}_v = \vec{w} \cdot \vec{s}_v^\alpha$ where $\vec{w} = (w_x, w_y, w_z) \in \mathbb{R}^3$ and $\vec{s}_v^\alpha = (\hat{s}_v^x, \hat{s}_v^y, \hat{s}_v^z)$. The scaling we have applied to the first term in $\hat{H}(\mathcal{G})$ means that its largest eigenvalue (by absolute value) will scale as $\mathcal{O}(L)$ and thus the energy per spin is always finite, independent of the choice of the graph.

The Hamiltonian $\hat{H}(\mathcal{G})$ encompasses a range of notable models of quantum magnetism and is a valid descriptor of non-magnetic systems such as fermions in the strongly interacting limit[23] or bosons with a maximum on-site occupancy[24]. Throughout this work we will supplement our analytical results with numerical calculations for the limit of $\hat{H}(\mathcal{G})$ which describes the competition between spin–spin correlations along the $z$ and $x$–$y$ spin-axes, respectively, i.e. where $\vec{w} = 0$ and $J_x = J_y$. We should emphasize that the coupling strengths in our Hamiltonian are isotropic, meaning our results do not apply to setups such as the SYK model, where the individual strengths are random and thus anisotropic.

### The average graph
To be as general as possible we will assume nothing about our simple graph $\mathcal{G}$ other than its size and draw it uniformly at random from the space of all simple graphs with $L$ vertices. Such a process is equivalent to drawing the graph from the Erdős–Rényi (ER) ensemble[25] with edges appearing independently with probability $p = 1/2$. We will reference an instance of a graph from the ER ensemble with edge probability $p$ as $\mathcal{G}_{ER}(p)$ and reference our Hamiltonian on this graph via $\hat{H}(\mathcal{G}_{ER}(p))$. In the case, $p = 1$ then $\mathcal{G}_{ER}(p)$ is equivalent to the complete graph $\mathcal{G}_{Complete}$—the simple, unweighted graph on $L$ vertices where all edges are present.

In order to determine the equilibrium physics of $\hat{H}(\mathcal{G}_{ER}(p))$ we focus on the structure of the two-body operators, $\sum_{(v,v')\in E} \hat{s}_v^\alpha \hat{s}_{v'}^\alpha$ in $\hat{H}(\mathcal{G}_{ER}(p))$. The problem of finding the eigenspectrum of these operators is equivalent to finding the number of edges cut for all possible partitions of the underlying graph into $2s + 1$ sets of vertices. On an ER graph with non-vanishing $p$ there exist strict bounds on this quantity and it cannot deviate significantly from its expected value[26,27]. In the Supplementary Information (SI) we utilise such observations to derive a strict bound on the maximum eigenvalue (by magnitude) of $\hat{H}(\mathcal{G}_{ER}(p)) - \hat{H}(\mathcal{G}_{Complete})$ and subsequently prove the following theorem:

**Theorem 1.** *Let $\mathcal{G}_{ER}(p)$ be an instance of the Erdős–Rényi graph with finite edge probability $0 < p \leq 1$ and $L$ vertices. Let $\mathcal{G}_{Complete}$ be the*

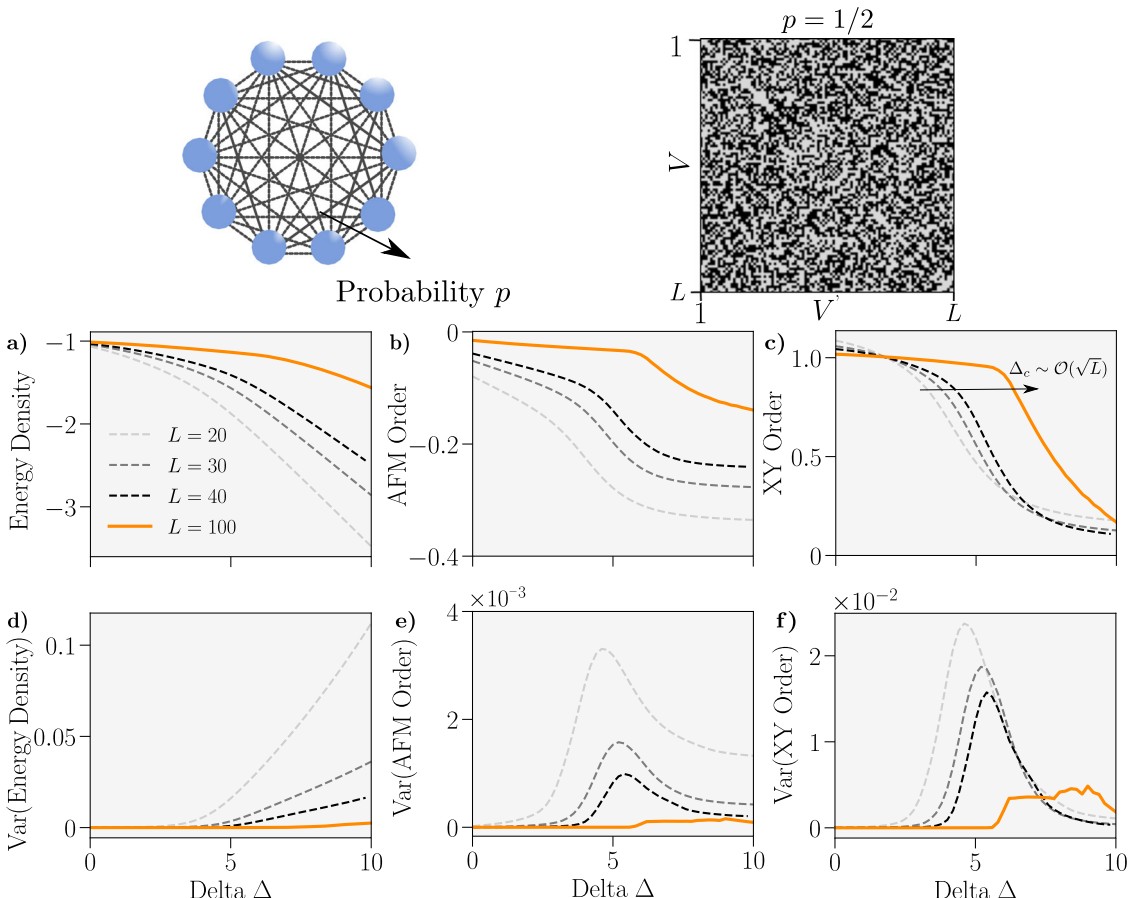

**Fig. 2 | Properties of the ground-state of the spin 1/2 *XXZ* Hamiltonian on the Erdős–Rényi (ER) graph.** A graph schematic is provided top left. We set $p = 1/2$ here and an example adjacency matrix for $L = 100$ is shown for this parameter in the top right. Data is obtained, for a given $\Delta$, by calculating the ground state for $n$ of draws of the ER graph from its ensemble and then averaging (plots **a**–**c**) or taking the variance (plots **d**–**f**). System sizes are coded by colour. **a**–**c** Energy density $\frac{1}{L}\langle H \rangle$, anti-ferromagnetic order $C_{\mathrm{AFM}}$ and *XY* order $C_{\mathrm{XY}}$—see Eq. (5)—versus $\Delta$. **d**–**f** Variance of the corresponding upper observables. We used $n = 100, 100, 100,$ and $10$ for $L = 20, 30, 40,$ and $100$ respectively.

complete graph on $L$ vertices. Define the free-energy density of a $d^L \times d^L$ matrix as $f(\hat{A}) = -\frac{1}{L\beta}\ln\left(\mathrm{Tr}(e^{-\beta \hat{A}})\right)$, where $\beta \in \mathbb{R}_{\geq 0}$ is the inverse temperature. Then, for a given spin $s$ and an arbitrary set of values for the microscopic parameters $\{J_x, J_y, J_z, w_x, w_y, w_z\}$,

$$\lim_{L \to \infty} f(\hat{H}(\mathcal{G}_{\mathrm{ER}}(p))) \equiv \lim_{L \to \infty} f(\hat{H}(\mathcal{G}_{\mathrm{Complete}})) \qquad \forall \beta \in \mathbb{R}_{\geq 0}, \qquad (2)$$

Moreover, for finite large $L$, we have $|f(\hat{H}(\mathcal{G}_{\mathrm{ER}}(p))) - f(\hat{H}(\mathcal{G}_{\mathrm{Complete}}))| = \mathcal{O}(L^{-1/2}) \ \forall \beta \in \mathbb{R}_{\geq 0}$.

From Theorem 1 it follows that all thermodynamic observables (i.e. ones that can be written as a function of the free energy density) are equivalent for the equilibrium states $\rho(\mathcal{G}_{\mathrm{ER}}(p))$ and $\rho(\mathcal{G}_{\mathrm{Complete}})$, with $\rho(\mathcal{G}) \propto \exp(-\beta\hat{H}(\mathcal{G}))$. The finite-size corrections, dictate that, for a single draw of the ER graph, the difference between such observables scales as $\mathcal{O}(L^{-1/2})$. Due to their decreasing nature, these statistical fluctuations about the average can be, with high probability, ignored for large $L$—with the limit $L \to \infty$ of $\mathcal{G}_{\mathrm{ER}}(p)$ being a single graph known as the Rado graph[28]. Thus when drawing a single graph from the space of all graphs (i.e. setting $p = 1/2$) the equilibrium properties of the system will, with increasing certainty and accuracy as $L$ increases, be equivalent to those of $\rho(\mathcal{G}_{\mathrm{Complete}})$. In the SI we also provide numerical calculations supporting the bound we derive on the maximum eigenvalue of $\hat{H}(\mathcal{G}_{\mathrm{ER}}(p)) - \hat{H}(\mathcal{G}_{\mathrm{Complete}})$.

To the best of our knowledge, this is the first proof of the equivalence between the ER and complete graph free energy densities for the general, quantum Hamiltonian in Eq. (1). Whilst such an

equivalence has been proven for the classical Ising model[29], our theorem encompasses this classical case and applies to a much broader range of Hamiltonians—both classical and quantum.

Theorem 1 leaves us with a remarkable conclusion: the equilibrium physics described by $\hat{H}(\mathcal{G})$, where $\mathcal{G}$ is sampled from the space of all simple graphs, is not many-body. This is because the free energy density as $L \to \infty$ is equivalent to that for a Hamiltonian built solely from the collective spin operators $\hat{S}^\alpha = \sum_v \hat{s}_v^\alpha$ with $\alpha = x, y$ or $z$. The eigenstates of such a Hamiltonian are collective, mean-field states of matter such as condensates and uniform product states.

We reinforce these results in Fig. 2, where we perform matrix product state (MPS) calculations of the ground state of the spin 1/2 *XXZ* Hamiltonian on the ER graph with $p = 1/2$. This Hamiltonian is explicitly defined in the Methods section, along with the order parameters $C_{\mathrm{XY}}$ and $C_{\mathrm{AFM}}$ for the *XY* and AFM (anti-ferromagnetic) phases, respectively. In the thermodynamic limit, we can use Theorem 1 to identify the Dicke state (see SI) $|\psi\rangle_{\mathrm{GS}} = |L/2, M\rangle$ with $M$ finite as the ground state independent of the $z$–$z$ interaction strength $\Delta > 0$.

In Fig. 2 we observe, explicitly, the convergence, as $L$ increases, of the ground-state to this condensate for all values of $\Delta$—in agreement with Theorem 1. Whilst on a finite-size system we observe a ground-state transition between the *XY* and AFM phases, we find that the critical point drifts and disappears as $L \to \infty$ due to an apparent scaling of $\Delta_c \sim \mathcal{O}(\sqrt{L})$ (see SI). We also calculate the variance of our observables (see the "Methods" section for definition) with respect to different draws of the graph from the ER ensemble. This variance decreases with

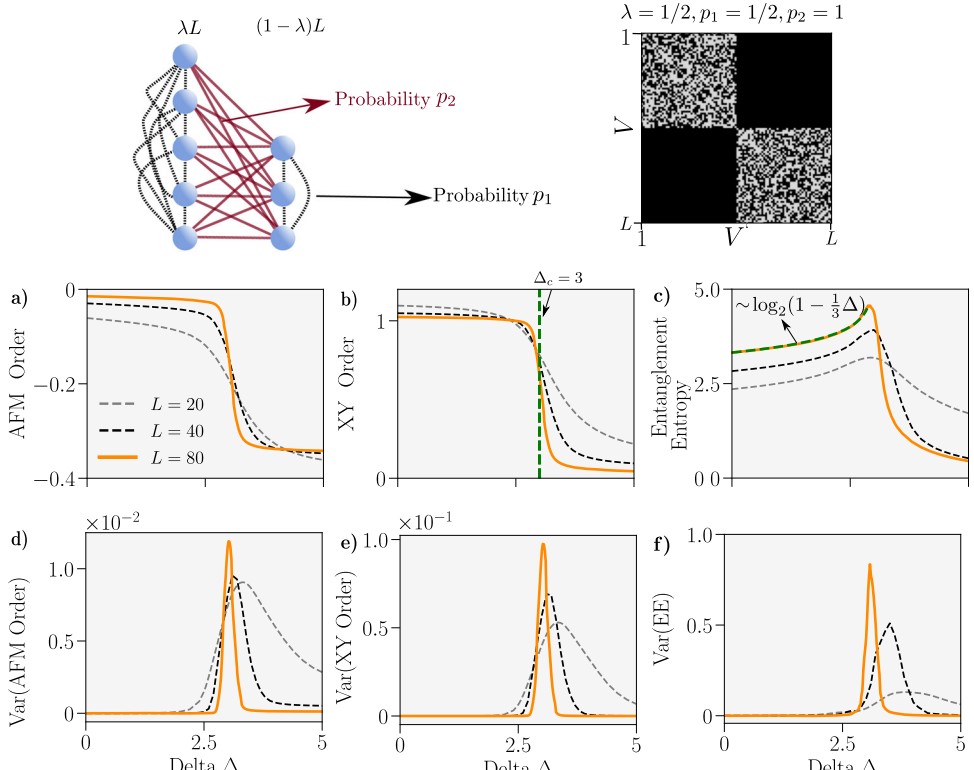

**Fig. 3 | Properties of the ground-state of the spin 1/2 *XXZ* Hamiltonian on a random graph with a non-trivial cut, $\mathcal{G}(\lambda,p_1,p_2)$.** A graph schematic is provided top left. We set $\lambda = 1/2$, $p_2 = 1$ and $p_2 = 1/2$ here and an example adjacency matrix for $L = 100$ and these parameters is shown top right. **a–c** Average anti-ferromagnetic order $C_{AFM}$, *XY* order $C_{XY}$ and Von-Neumann entanglement entropy (EE) between the two partitions versus $\Delta$ for graphs drawn $n$ times from $\mathcal{G}(\lambda,p_1,p_2)$. A curve of the form $-0.265\log_2(1-\frac{1}{3}\Delta) + \text{EE}(\Delta = 0)$ is fitted to the entanglement entropy for $\Delta < 3$ and $L = 80$ and marked in green. The line $\Delta_c = 3$, which corresponds to the prediction of Eq. (3), has been marked in green on the plot of the *XY* order. **d–f** Variance of the corresponding upper observables. We used $n = 100$ for all values of $L$.

$L$, indicating that the ground-state properties are converging to a single fixed limit for $L \to \infty$, where we can make statements about them with certainty. In the SI we use MPS methods to also demonstrate this convergence for the ground state of the Transverse Field Ising Model on an ER graph.

It is clear then that one must therefore pick exceptional graphs—i.e. graphs from subspaces of the space of all graphs which are vanishingly small with respect to the full space—in order to witness more complex, strongly-correlated states. These exceptions are graphs with some non-trivial cut where, even as $L \to \infty$, the number of edges between partitions deviates significantly from its expected value. We will now consider such graphs.

## Graphs with a non-trivial cut

We take a simple graph of $L$ sites along with a bi-partition of these sites into two sets $A$ and $B$ comprising $\lambda L$ and $(1-\lambda)L$ sites respectively ($0 < \lambda < 1/2$). We assume nothing other than that the ratio $\alpha$ of the number of edges between sites in different sets to the number of edges in the whole graph $N_E$ differs from its expected value of $2\lambda(1-\lambda)$, even as $L \to \infty$. We reference a graph constructed uniformly at random from this ensemble as $\mathcal{G}(\lambda,p_1,p_2)$, where $p_1$ and $p_2$ are finite numbers that can be directly related to $\alpha$ and $N_E(\mathcal{G}(\lambda,p_1,p_2))$ (see the "Methods" section).

In the SI we prove a theorem that dictates that the free energy density for $\hat{H}(\mathcal{G}(\lambda,p_1,p_2))$ is equivalent, in the thermodynamic limit, to that of an effective Hamiltonian which is built solely from the collective operators $\hat{S}_A^\alpha = \sum_{v \in A} \hat{s}_v^\alpha$ and $\hat{S}_B^\alpha = \sum_{v \in B} \hat{s}_v^\alpha$. This leads to another striking conclusion: if one were to randomly select a graph from those with a non-trivial cut they would find, as $L$ increases, that the system realised by $\hat{H}(\mathcal{G})$ is increasingly similar to that of a pair of interacting collective spins. Whilst the physics of such a system is richer than that for the ER ensemble, it is still collective and not many-body.

In Fig. 3 we apply our MPS calculations to the $s = 1/2$ *XXZ* limit of $\hat{H}(\mathcal{G}(\lambda,p_1,p_2))$. We directly observe an emergent first-order transition between an *XY* phase and an AFM phase. The properties of the ground state in the *XY* regime demonstrate convergence towards the condensate $|L/2,0\rangle$ whilst, in the AFM regime, the properties are that of the product state where the spins in the sets $A$ and $B$ polarise in opposite directions along the spin-$z$-axis. These results are in agreement with our reduction of the system to a pair of collective spins, which predicts (see SI) a critical point of

$$\Delta_c = \frac{p_1(\lambda^2 + (1-\lambda)^2) + 2p_2\lambda(1-\lambda)}{2p_2\lambda(1-\lambda) - p_1(\lambda^2 + (1-\lambda)^2)}. \tag{3}$$

Again, we are averaging the ground-state properties over draws of the graph from its corresponding ensemble. With increasing $L$ the peak in the variance appears to be tending towards a delta function centered on $\Delta_c$, where we do not expect the variance to vanish due to the first-order discontinuity.

What is the key property of the graphs $\mathcal{G}_{ER}(p)$ and $\mathcal{G}(\lambda,p_1,p_2)$ which allows the equilibrium physics of $\hat{H}(\mathcal{G})$ to be reducible to a collective spin model? Despite the fact they are not invariant under a given permutation of two spins, our proof tells us that the statistical fluctuations which break these symmetry fluctuations cannot affect the thermodynamic properties of the system. As a result, we can ignore them as $L \to \infty$ and work with a simpler Hamiltonian built solely from collective spin operators. The equilibrium properties of the system can

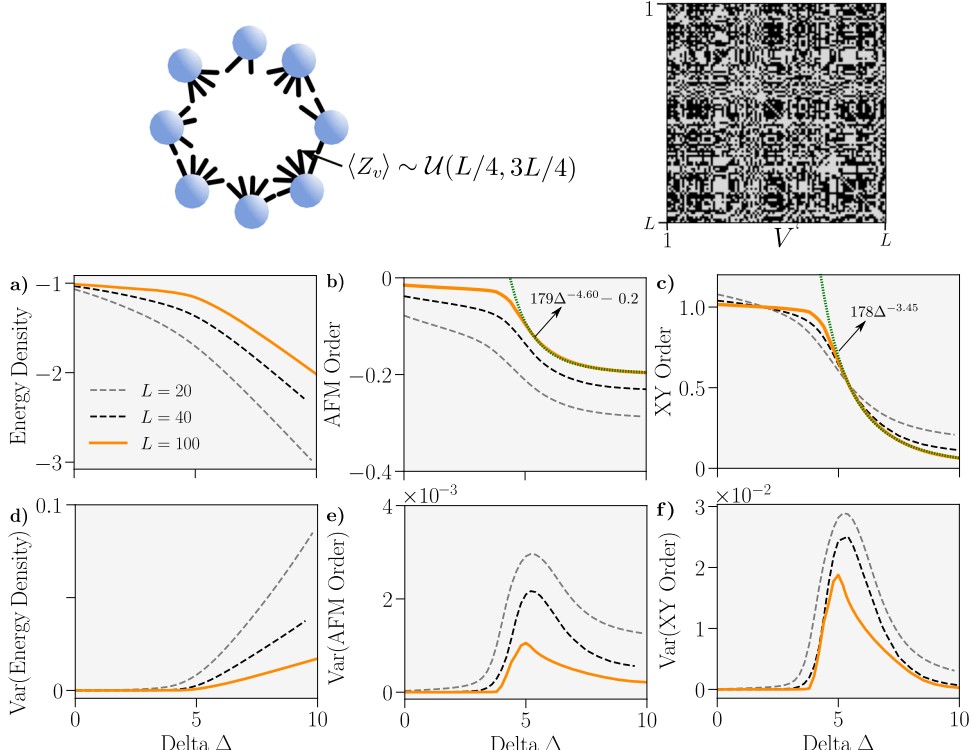

**Fig. 4 | Properties of the ground-state of the spin 1/2 *XXZ* Hamiltonian on the graph where each site has a degree $Z_V$ drawn from $\mathcal{U}(L/4, 3L/4)$, the uniform discrete distribution over the interval $[L/4, 3L/4]$.** Graph schematic alongside adjacency matrix for $L = 100$ is provided. **a–c** Average energy density, anti-ferromagnetic order $C_{AFM}$ and *XY* order $C_{XY}$ versus $\Delta$ for graphs drawn $n$ times from the ensemble. Power-law curves have been fitted to the XY and AFM orders for $\Delta > 2$ and marked in green. **d–f** Variance of the corresponding upper observables. We used $n = 100, 100$ and $10$ for $L = 20, 40$ and $100$, respectively.

then be calculated with a semi-classical solution. It is worth emphasizing that such a solution does not capture the quantum fluctuations which are included in our numerics and present in the true ground state in the form of non-zero entanglement entropy and simultaneous long-range order along the $x$ and $y$ spin axes.

These results necessitate searching for exceptional structures where complex, truly many-body physics is manifest. Sparse, regular structures like the hypercubic lattice are the well-known exception as our proofs are reliant on the graph being dense (i.e. $N_E/L^2$ cannot vanish as $L \to \infty$). Theoretical results for various limits of $\hat{H}(\mathcal{G})$ on such structures are numerous and their capacity to host complex, many-body states of matter is well established[30–34]. Despite being so strongly distinct from the average case, such structures are ubiquitous in nature. Our results suggest that if these exceptional structures were not commonplace, the world around us would not be able to exhibit such complex, rich behaviour.

**Irregular dense graphs**

Importantly, within the space of dense graphs we are able to discover hitherto unknown exceptional structures which can host complex many-body phases of matter. As far as we are aware, such structures have never before been treated in the realm of many-body physics. Specifically, in the following, we consider 'irregular dense graphs' where, even as $L \to \infty$, we cannot split the sites into a finite number of sets where sites in the same set have identical values of $Z_V/L$, with $Z_V$ the coordination number of a given site $V$. Such a statement is not true of $\mathcal{G}_{ER}(p)$ and $\mathcal{G}(\lambda, p_1, p_2)$.

As a prototypical example of an irregular dense graph, we consider one drawn from the ensemble of graphs where the degree of each site is assigned a value uniformly in the range $[L/4, 3L/4]$. Details on how we construct an instance of this graph are given in the "Methods".

We perform MPS calculations to find the ground state of the *XXZ* model on this inhomogeneous structure, with the results pictured in Fig. 4. Again, the general reduction in the variance of observables as $L$ increases suggests that there is a well-defined thermodynamic limit for this ensemble, with an emergent continuous phase transition between the *XY* and AFM phases. We find, by numerical fitting, the associated order parameters follow a power law for $\Delta > \Delta_c \approx 2$.

We expand on these results further by considering, in Fig. 5, the 'maximally irregular' graph where every site, other than one pair of sites by necessity, has a different degree. We observe an emergent second-order phase transition on this graph with the same functional scaling of the order parameters as in Fig. 4. Unlike the other graphs considered in this work, the ground state of the system does not always possess $\langle \hat{S}^z \rangle = 0$ and its variation with $\Delta$ is responsible for the discrete changes in the observables in Fig. 5.

As a measure of the complex, inhomogeneous nature of the ground state on the maximally irregular graph, we calculate the Shannon entropy (SE) of the matrices of two-point $z$–$z$ and $x$–$x$ correlations (see the "Methods" section for definition) as a function of $\Delta$. These matrices can be viewed as images of the correlations in the system with each two-point correlation constituting a 'pixel' whose value is bounded between 1 and −1. The SE of this image then specifies the amount of information contained within it and is a widely used measure in image reconstruction and classification algorithms[35,36]. Here we apply this measure to the image of correlations in a quantum system, noting that it vanishes if all two-point correlations are the same (such as in the Dicke state $|L/2, M\rangle$) and is maximised when the correlations are distributed uniformly across $[-1, 1]$.

In Fig. 5 we observe a sharp increase in the SE for the $x$ and $z$ correlations at the critical point, with the $z$–$z$ correlations close to maximising this entropy and meaning that the corresponding

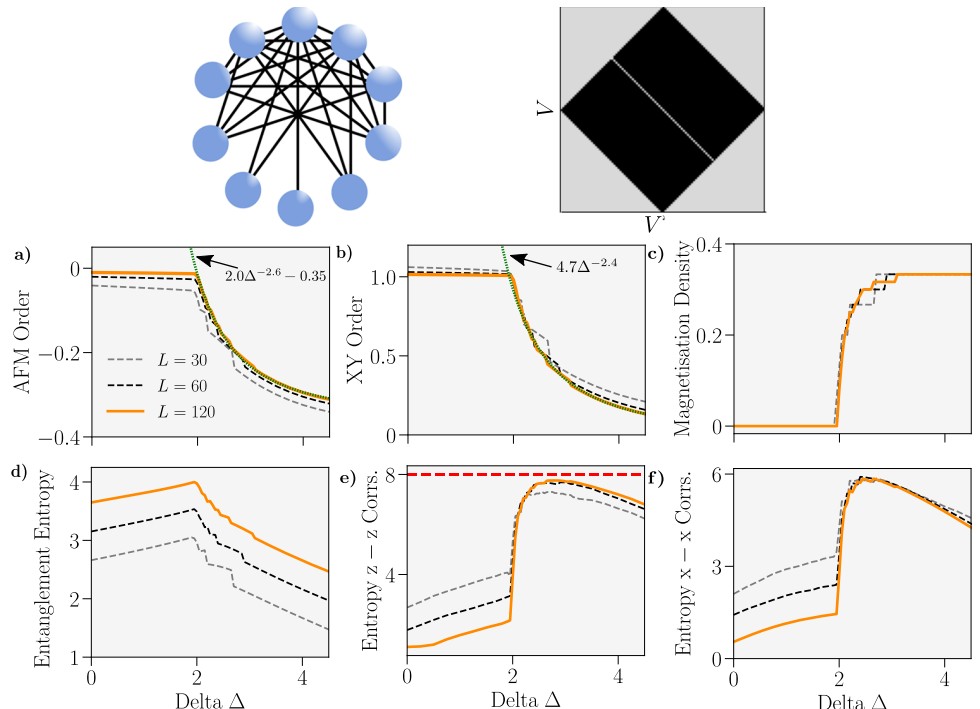

**Fig. 5 | Properties of the ground-state of the spin 1/2 *XXZ* Hamiltonian on the 'maximally irregular graph' where each pair of vertices, other than two, have a different degree.** Graph schematic for $L = 10$ alongside adjacency matrix for $L = 100$ is provided. System sizes are coded by colour. **a–f** Anti-ferromagnetic order $C_{AFM}$, *XY* order $C_{XY}$, magnetisation density $\langle \hat{S}^z \rangle / L$, von-Neumann entanglement entropy between sites $V = 1 \ldots L/2$ and $V = L/2 + 1 \ldots L$ and Shannon-Entropy of the $z$–$z$ and $x$–$x$ correlations (see the "Methods" section) versus $\Delta$. We use $n = 256$ bins to calculate the Shannon Entropy. Power-law curves have been fitted to $C_{AFM}$ and $C_{XY}$ for $\Delta > 2$ and $L = 120$ and marked in green. A dotted red line has been added to plot **e)** to indicate the maximum possible value for the Shannon Entropy with the number of bins used.

correlation matrix cannot be compressed without a loss of information[37]. The phase transition in the maximally irregular graph is thus heralded by a vast amount of complexity in the system's correlations and along all spin-axes. In the SI we compare this entropy for the graphs considered in this work as well as the 1D chain. The entropy is largest for the irregular dense graphs and these are the only ones where it does not diminish with system size and is non-zero simultaneously in the $x$, $y$ and $z$ degrees of freedom as $L \to \infty$. We also perform calculations for the Transverse Field Ising limit of $\hat{H}(\mathcal{G})$ on the maximally irregular graph and observe similar behaviour in this entropy as $L \to \infty$ in the vicinity of the critical point.

Unlike other existing measures of complexity in quantum states such as the average disparity[38], features like translational invariance are accounted for by our measure and reduce the informational complexity associated with a quantum state. Given that there is no disorder or anisotropy in the microscopic parameters of our Hamiltonian, such results warrant our interpretation of irregular dense graphs as a new class of geometries on which many-body systems can exhibit novel, complex phases of matter.

## Discussion

Our results open up a number of promising avenues for future research. Numerical methods in Condensed Matter physics have been optimised over the last few decades to treat quantum systems hosted on regular, sparse, translationally invariant structures. Our work instigates the need to alter existing methods, in order to account for more complex, inhomogeneous geometries and allow us to probe the physics, both in and out-of-equilibrium, of such systems.

In an equilibrium context, it would be important to understand how geometric irregularity affects the properties of various Hamiltonians—including those with anisotropic, random couplings[17]. In an out-of-equilibrium setting, a number of questions arise about how exotic

phenomena such as time-crystalline order[39,40], quantum synchronisation[41,42], and heating-induced order[43] are affected in the presence of such geometric irregularity. These results could enable the engineering of quantum states of matter in many-body simulators with new, geometrically enabled functionalities.

## Methods

For our numerical results, we used state-of-the-art Matrix Product State (MPS) calculations for the *XXZ* $s = 1/2$ limit of Eq. (1). The Hamiltonian in this limit reads

$$H_{XXZ}(\mathcal{G}) = \frac{L}{N_E} \sum_{(v,v') \in E} -J(\hat{\sigma}_v^x \hat{\sigma}_{v'}^x + \hat{\sigma}_v^y \hat{\sigma}_{v'}^y) + \Delta \hat{\sigma}_v^z \hat{\sigma}_{v'}^z, \qquad (4)$$

which corresponds to setting, in Eq. (1), $J_x = J_y = -J$, $J_z = \Delta$, $\vec{w} = 0$ and $\hat{s}_v^\alpha = \hat{\sigma}_v^\alpha$—dropping the factor of 1/2 in front of the Pauli matrix $\hat{\sigma}_v^\alpha$ for simplicity. We restrict ourselves to $\Delta \geq 0$, set $J = 1$ and define order parameters for the *XY* and anti-ferromagnetic (AFM) phases, respectively, via

$$C_{AFM} = \frac{1}{N_E} \sum_{(v,v') \in E} \langle \hat{\sigma}_v^z \hat{\sigma}_{v'}^z \rangle,$$

$$C_{XY} = \frac{1}{N_E} \sum_{\substack{v,v'=1 \\ v > v'}}^{L} \langle \hat{\sigma}_v^x \hat{\sigma}_{v'}^x + \hat{\sigma}_v^y \hat{\sigma}_{v'}^y \rangle. \qquad (5)$$

These take a non-zero value in their respective phases, vanish in the opposing phase and, importantly, can be calculated for the ground state on any graph. The XXZ Hamiltonian has rotational symmetry around the spin-$z$-axis meaning $\langle \hat{\sigma}_v^x \hat{\sigma}_{v'}^x \rangle = \langle \hat{\sigma}_v^y \hat{\sigma}_{v'}^y \rangle$ for the ground state. The parameter $C_{XY}$ thus quantifies the simultaneous off-diagonal order in the $x$ and $y$ degrees of freedom.

In this work, we also introduce the Shannon Entropy of the spin-correlations in a given state as

$$H\left(\langle \hat{\sigma}_v^\alpha \hat{\sigma}_{v'}^\alpha \rangle\right) = \sum_{i=0}^{n-1} p_i \log_2(p_i), \qquad (6)$$

where $p_i$ is the fraction of elements of the $L \times L$ matrix $\langle \hat{\sigma}_v^\alpha \hat{\sigma}_{v'}^\alpha \rangle$ which are between $-1 + 2i/n$ and $-1 + 2(i+1)/n$. The integer $n$ is the number of bins used to 'bin up' the matrix elements. We use $n = 256$ throughout in order to make the connection between $\langle \hat{\sigma}_v^\alpha \hat{\sigma}_{v'}^\alpha \rangle$ and a grayscale image of the correlations in the system. This entropy measure is bounded as $0 \le H(\langle \hat{\sigma}_v^\alpha \hat{\sigma}_{v'}^\alpha \rangle) \le \log_2(n)$ and can be interpreted as the amount of information required to encode the distribution or image of off-diagonal correlations in a state.

Our DMRG calculations often involve ensemble-averaging the ground state properties over a series of random graphs drawn from a corresponding ensemble for a given $\Delta$ and $L$. In order to assess the convergence of the order parameters $C_{XY}$ and $C_{AFM}$ to a well-defined thermodynamic limit we calculate their variances $\text{Var}(C_{XY})$ and $\text{Var}(C_{AFM})$. These are defined as

$$\text{Var}(C) = \frac{1}{n-1}\sum_{i=1}^{n}(C^i - \bar{C})^2, \qquad (7)$$

for $n$ draws of the graph from its ensemble and where $\bar{C}$ is the average of the ground state order parameter for these different draws. For a given $L$ and $\Delta$, the variance then tells us the fluctuation in the ground state properties when averaging over random instances of the given graph ensemble. As the order parameters have been appropriately normalised by the size of the graph the variance will allow us to infer the convergence of the ground state properties in the thermodynamic limit. We discuss explicit details of our matrix product state calculations and implementation in the Supplementary Information. We also provide an analysis of the truncation errors and energy convergence involved in our DMRG calculations. The ground state in our calculations typically has a bipartite entanglement entropy that scales logarithmically with the partition size and makes it tractable using a bond dimension proportional to the system size. Such a scaling has previously been observed in the Lipkin–Meshov–Glick model[44] (an all-to-all $XY$ model with a transverse field) and here we observe it for the $XXZ$ model on a range of dense graphs.

In the main text, we introduced the non-trivial cut graph where we assumed nothing but that there exists some bi-partition of the $L$ vertices into two sets, size $\lambda L$ and $(1-\lambda)L$, and where the cut-size $\alpha$ (i.e. the ratio of the number of edges between the two sets to the total number of edges in the graph $N_E$) is not equal to its expected value $2\lambda(1-\lambda)$. We can construct such a graph by taking $L$ sites, partitioning them into the two corresponding sets, and randomly assigning edges between sites in the same set with probability $p_1$ and between sites in different sets with probability $p_2$. The values of $\alpha$ and $N_E$ are then approximately

$$N_E \approx p_1(\lambda^2 - \lambda + 1/2) + p_2\lambda(1-\lambda)L^2,$$
$$\alpha \approx \frac{2p_2\lambda(1-\lambda)}{p_1\lambda^2 + p_1(1-\lambda)^2 + 2p_2\lambda(1-\lambda)}, \qquad (8)$$

which becomes exact as $L \to \infty$, with each unique pair of values of $p_1$ and $p_2$ uniquely specifying $N_E$ and $\alpha$. As a result, the parameters $p_1$ and $p_2$ can be interchanged freely with $\alpha$ and $N_E$ in this limit and lead us to define an instance of this graph as $\mathcal{G}(\lambda, p_1, p_2)$. Provided $p_1 \ne p_2$ and $0 < \lambda < 1$ then we find $\alpha \ne 2\lambda(1-\lambda)$, meaning $\mathcal{G}(\lambda, p_1, p_2)$ has a non-trivial cut. Our construction routine—given its independent treatment of edges—will uniformly sample over all graphs with $L$ vertices, a number of edges $N_E$ and non-trivial cut-size $\alpha$.

We also introduced the uniform variate graph where the degree of each site follows the discrete uniform distribution $\mathcal{U}(L/4, 3L/4)$. We have used lower and upper bounds sufficiently separated from 0 and $L - 1$ to avoid creating non-graphical degree distributions. To generate a given graph from this ensemble, we draw the degree distribution by randomly generating the degree of each site—repeating until a sum of the degrees is even—and then using the Havel Hakimi (HH)[45] algorithm to generate a graph with the given degree distribution. Such a routine does not sample uniformly from the space of all graphs with degrees drawn from the distribution $\mathcal{U}(L/4, 3L/4)$ due to the high assortativity[46] bias in the HH algorithm. We are, however, unaware of an algorithm that can generate unbiased samples of dense, inhomogeneous graphs with a given degree distribution. Hence, our results are for the ensemble of graphs with degrees drawn from the distribution $\mathcal{U}(L/4, 3L/4)$ and generated by the HH algorithm.

## Data availability
The data that was used to create the plots within this paper is provided as source data .

## Code availability
The Network Python (TeNPy) library[47], which can be used to perform the simulations in the article, is available at https://tenpy.readthedocs.io/en/latest/. The programming scripts used to obtain the source data in this manuscript are available from the corresponding author upon reasonable request.

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

## Acknowledgements

The simulations used to produce the computational results of our work were run on the University of Oxford Advanced Research Computing (ARC) facility and involved over 100,000 h of exclusive use of nodes each equipped a with 48-core 2.9 GHz Intel Cascade Lake Processor. J.T. is grateful for ongoing support through the Flatiron Institute, a division of the Simons Foundation. D.J. and J.T. acknowledge funding from EPSRC grant EP/P009565/1. D.J. also acknowledges funding from the Cluster of Excellence 'Advanced Imaging of Matter' of the Deutsche Forschungsgemeinschaft (DFG)—EXC 2056—project ID 390715994. A.S. acknowledges funding from the UK Engineering and Physical Sciences Research Council as well as from the Smith-Westlake scholarship.

## Author contributions

J.T. provided the idea for the project, formulated the proofs and wrote the manuscript. J.T., A.S. and A.A. ran the numerical simulations and analysed the data. J.T. and D.J. edited the manuscript with help from A.S. and A.A. All authors contributed substantially to discussions on the results and ideas contained therein. D.J. oversaw and managed the overall project.

## Competing interests

The authors declare no competing interests.
