## [Peer Review File · Nature Communications]

REVIEWER COMMENTS

Reviewer #1 (Remarks to the Author):

This was a very interesting paper. I am open to it being published if my comments can be addressed satisfactorily. However, some of the claims that are made were misleading and I think the result is not quite as grand as the authors had originally claimed. I believe with better comparisons to other work and a sharper notion of when many-body physics appears collective, this will be a very improved paper. Below I list my main comments and concerns:

1) Eq. (1) is a very special type of Hamiltonian. It is not too surprising that many-body physics might disappear in favor of emergent single-spin physics, if we start from a complete graph and slowly remove edges. (It is of course interesting to quantify how many edges can be removed...as this paper does.). However this is not the only type of model on a complete graph. For example the S-K spin glass, or more recently the SYK model or SY type models, are also on all-to-all connected graphs and definitely are not describable by an emergent single spin system. Therefore, the authors need to emphasize that their conclusions are quite special to the particular Hamiltonian in Eq. (1).

There is another feature of this type of Hamiltonian which is likely quite special. I believe that in a typical state, it is likely that the energy of $H(G)$ is very close to 0 -- so the thermodynamics of these models are already visibly peculiar -- they will be dominated by an anomalously small number of states relative to a finite-dimensional system. It may be worth commenting on this point in the paper.

2) I think it is misleading to say that the Hamiltonian acts on a Hilbert space of an effectively small dimension $2sL+1$. The Hamiltonian acts on a Hilbert space with a fixed dimension of $(2s+1)^L$. What I assume happens, and the authors should clarify, is that the Hamiltonian is approximately extremely degenerate, and the Hilbert space can be split into (exponentially many) irreps of $SU(2)$.

In the main text, it sounds to me as though the authors are focusing mainly on the unique spin- s^*L irrep. I don't entirely understand why no other ones are interesting -- for example, there might be a handful (2?) s^*L-1 dimensional irreps? Can the authors more carefully explain what happens to the spectrum in one of these irreps? I would imagine the ground state may indeed always be in the s^*L irrep, but are there no low-lying levels in any other irrep?

On a complete graph this problem must be exactly solvable I'd imagine, and so the authors can carefully track exactly what happens, and then compare to the ER graph.

3) I am confused by the discussion on page 2 of the supplement. I worry that the authors are not adequately considering the quantum nature of the operators M_α , which are simply the generators of an $SU(2)$ symmetry (when $\beta's = 0$). The α -bases in (S6) etc. are different for each α -- asymptotically at large L one can diagonalize at leading order in L all spin operators simultaneously, but this is an approximation and will fail at any finite L . As per point 1), likely most of the spectrum of these models is near $H \approx 0$ and it may be that for typical states, the approximation that all M_α 's can be simultaneously diagonalized (which seems to me required to follow the logic on page 2) is not valid.

I think the authors need to address this point (which also related to point 2 above) quite carefully in revisions. In particular, the authors need to clarify whether their discussion in supplemental section 1 is valid only in the s^*L -irrep's Hilbert space, but not more generally. Towards this end, more words or calculations need to be provided to better elucidate why it is acceptable to consider different vectors in different bases when evaluating each term in $H|\sigma\rangle = E|\sigma\rangle$.

4) Lastly, I remark that for plenty of the interesting dynamical applications of this interesting family of Hamiltonians, it may be very important to not only study the ground state, but also highly excited states, where I am concerned the analysis of this paper does not hold.

5) If the authors believe my claims about typical vs. ground states are incorrect, I will need to see a more careful analysis of the full spectrum of the Hamiltonian for some small system sizes. In fact, this may be a good idea to explore anyway in a new supplemental appendix. Is the spectrum extremely degenerate near $H=0$? Does random matrix statistics hold for the middle 20 percent of all energy levels? The method proposed in cond-mat/0610854 is quite numerically efficient and probably would work well to diagnose this question. If GUE/GOE etc. statistics are observed, this suggests that much of the spectrum is chaotic and not simple/single-particle.

Reviewer #2 (Remarks to the Author):

The paper considers spin models on a class of graphs with a high connectivity (dense graphs): each site is connected to a number of sites that scales proportionally to the total number of sites L . The authors focus on a spin-1/2 model with competing interactions: ferromagnetic in XY plane and

antiferromagnetic for Z components. Their central statement is that, within this class of graphs, there is a subset of irregular dense graphs that exhibit interesting physics (complexity) that is not found in more conventional, regular dense graphs.

The authors first consider Erdős-Renyi graphs, where each link is present with probability $1/2$ (i.e. each site is connected to $L/2$ sites, up to \sqrt{L} fluctuations), Fig. 2. Not surprisingly, they find a mean-field behavior in this situation. They then modify this model, by splitting the whole set of spins in two subsets, with different probabilities of the edge within the subset and between the subsets. Again, the numerical (DMRG) results, Fig. 3, are in agreement with conventional mean-field expectations. After this, the central part of the paper comes. The authors consider the model on an “irregular dense graph” — with a connectivity that is still in average proportional to L but varies substantially between the nodes. Specifically, the connectivity is first chosen to be in the range $[L/4, 3L/4]$ (Fig. 4) and then $[1, L-1]$ (“maximally irregular graph”, Fig.5). The authors argue that this subclass of graphs lead to a new physics that have not been identified previously. The key evidence provided by the authors in favor of this is the behavior of the Shannon entropy describing the complexity of the distribution of spin correlations over all pairs of sites (v, v') , which is presented in Fig. 5.

The paper is interesting, and I can imagine that the class of graphs identified in this work may attract attention of other researchers. While these results deserve to be published, I am uncertain at this stage concerning a recommendation for publication in Nature Communications. The main claim of the authors is essentially novel and important features of the models that they have identified, and I am not fully convinced. The main argument, as I understood it, is that in Fig. 5e and 5f the Shannon entropy almost reaches its maximal value at Δ_c . The authors argue that this is essentially different from the behavior for more conventional dense graphs, as shown in Fig. S3 of the Supplementary Material. At the same time, if one looks at the ER graph (left panels) in Fig. S3, the behavior is rather similar: nearly maximal value of the Shannon entropy close to Δ_c , with the difference that Δ_c scales with L (as a \sqrt{L}). This behavior of Δ_c for ER graphs is easy to understand in the considered model (the mean field solution favors ferromagnetism rather than antiferromagnetism for a finite Δ at large L); it can be different for other phase transitions. So, I am not persuaded by the degree of innovation and importance with respect to models on the graphs introduced in the paper. Can the authors make this more convincing?

A few additional questions / comments:

1) In Fig. 4b and 4c, there are fits by power laws with some exponents (4.6 and 3.45). Do the authors propose that these fits should be taken seriously in the sense of asymptotic behavior? Are there analytical arguments in favor of such a behavior? What is then expected for the exponents? Or is this just a numerical observation?

2) Similar questions with respect to Fig. 5a and 5b.

3) Minor comment (I suppose, just typos):

In Fig. 4b, in the fit, the constant +0.2 should be apparently -0.2 (minus sign)

And, similarly, in the fit in Fig. 5a, +0.35 should be apparently -0.35

4) Fig. 5 caption:

“... where each pair of vertices, other than two, have a different degree.”

Should this be “where each vertex ... has a different degree” ?

5) In Supplementary Information, concerning the convergence of the DMRG calculation with MPS:

“The truncation error is never greater than 1.2×10^{-4} . Combining this with the smoothness of the observables as a function of Δ suggests our results are reasonably accurate and well-converged.”

To demonstrate convergence more convincingly, it would be natural if the authors provide results for several values of the bond dimension (the maximal one used for the calculation and a couple of values smaller by a factor ~ 2).

Dear Editor and Referees,

We thank both referees for their thorough reading and comments on our article. Both referees found our work 'interesting' and likely to attract the attention of other researchers. Referee 1 is open to our paper being published in Nature Communications following satisfactory revisions and addressal of their comments, whilst Referee 2 is 'unsure' and would like us to address their concerns about the degree of innovation.

In the following document we address the referees' comments and the concerns of referee 2 on a point-by-point basis. We then provide a list of changes we have made to the manuscript based on their comments.

We believe that, following these revisions, the clarity of our manuscript has been improved significantly and its impact to the community is clear. We hope that it is suitable for publication in Nature Communications in its present form.

Response to Referee Comments

Referee 1

Comment 1

Eq. (1) is a very special type of Hamiltonian. It is not too surprising that many-body physics might disappear in favor of emergent single-spin physics, if we start from a complete graph and slowly remove edges. (It is of course interesting to quantify how many edges can be removed...as this paper does.). However this is not the only type of model on a complete graph. For example the S-K spin glass, or more recently the SYK model or SY type models, are also on all-to-all connected graphs and definitely are not describable by an emergent single spin system. Therefore, the authors need to emphasize that their conclusions are quite special to the particular Hamiltonian in Eq. (1).

Response

We agree that the isotropy of the coupling strengths in our Hamiltonian is a feature that makes the Physics different to models with anisotropic couplings which originate from randomness such as in the SYK model. We have emphasized this in our definition of the model and also mentioned in the conclusion that studying random-coupling models on the graphs we consider would be an interesting area of future study.

Comments 2 + 3

2) I think it is misleading to say that the Hamiltonian acts on a Hilbert space of an effectively small dimension $2sL+1$. The Hamiltonian acts on a Hilbert space with a fixed dimension of $(2s+1)^L$. What I assume happens, and the authors should clarify, is that the Hamiltonian is approximately extremely degenerate, and the Hilbert space can be split into (exponentially many) irreps of $SU(2)$.

*In the main text, it sounds to me as though the authors are focusing mainly on the unique spin- s^*L irrep. I don't entirely understand why no other ones are interesting -- for example, there might be a handful (2?) s^*L-1 dimensional irreps? Can the authors more carefully explain what*

*happens to the spectrum in one of these irreps? I would imagine the ground state may indeed always be in the s^*L irrep, but are there no low-lying levels in any other irrep?*

On a complete graph this problem must be exactly solvable I'd imagine, and so the authors can carefully track exactly what happens, and then compare to the ER graph.

3) I am confused by the discussion on page 2 of the supplement. I worry that the authors are not adequately considering the quantum nature of the operators M_α , which are simply the generators of an $SU(2)$ symmetry (when $\beta = 0$). The α -bases in (S6) etc. are different for each α -- asymptotically at large L one can diagonalize at leading order in L all spin operators simultaneously, but this is an approximation and will fail at any finite L . As per point 1), likely most of the spectrum of these models is near $H \approx 0$ and it may be that for typical states, the approximation that all M_α 's can be simultaneously diagonalized (which seems to me required to follow the logic on page 2) is not valid.

*I think the authors need to address this point (which also related to point 2 above) quite carefully in revisions. In particular, the authors need to clarify whether their discussion in supplemental section 1 is valid only in the s^*L -irrep's Hilbert space, but not more generally. Towards this end, more words or calculations need to be provided to better elucidate why it is acceptable to consider different vectors in different bases when evaluating each term in $\langle H | \sigma \rangle = \langle E | \sigma \rangle$.*

Response

The Hilbert space is still exponentially large but the exponentially many degeneracies mean the number of relevant degrees of freedom can be reduced dramatically without a loss of information. We have clarified this.

Our results concerning the Erdős–Rényi graph in the thermodynamic limit are exact and valid for the spectrum of the whole Hamiltonian and not just the low-lying manifold or a specific $SU(2)$ irreducible representation. The true ground state of this Hamiltonian, in the XXZ case, will always lie in the irrep with spin sL and hence why we mention it. There will be other low-lying states in other irreps (such as the one mentioned by the referee) but as our numerical results focused solely on the ground-state we did not mention these. As we discuss further in the subsequent comment, our proof implies equivalence of thermodynamic quantities for any equilibrium state and not just the ground state.

We have made significant changes to the manuscript in order to make this much clearer and that our results apply to the full spectrum of $H(G_{ER}(p))$ and are not just specific to a certain $SU(2)$ irrep. This includes: i) extra discussion, ii) more explicit details and equations in our proof, iii) numerics demonstrating the convergence of $\langle H(G_{ER}(p)) \rangle$ and $\langle H(G_{Complete}) \rangle$ for high energy states in various different $SU(2)$ irreps and iv) numerical results for $H(G)$ on the Transverse Field Ising Model.

We note the objects M_α we introduced in the SM are not operators but real numbers representing the total spin of a given eigenstate of $O^\alpha(G)$. The reason that we are able to treat the O^α s separately is because for each α , we have proved the operator $O^\alpha(G_{ER}(p))$ is identical to its complete graph counterpart in the thermodynamic limit. The sum of the operators is thus also identical to its complete graph counterpart and we thus do not ever need to be concerned about simultaneously diagonalizing these operators. As we are only creating a linear combination of a

finite number (three) of such operators our statement about their finite-size deviations from the complete graph are also valid provided the linear coefficients are finite. We have added several equations and points of discussion in Section 1 of the SM to make this much clearer. We have also introduced hat notation for all operators throughout the manuscript, so it is obvious what is a scalar and what isn't.

Comment

Lastly, I remark that for plenty of the interesting dynamical applications of this interesting family of Hamiltonians, it may be very important to not only study the ground state, but also highly excited states, where I am concerned the analysis of this paper does not hold.

Response

Our results prove the equivalence of our Hamiltonian (which physically corresponds to the energy density of the system) on the Erdős–Rényi and Complete graphs (see points above). Their free energy densities at any temperature are thus equivalent, meaning the equilibrium value of any thermodynamic observable will be identical on both graphs. We have added a section to the SM (titled ‘*Normalisation and physical meaning of $H(G)$* ’) specifically discussing this and also emphasized it in the main text. We do not make any statements about out-of-equilibrium behavior as this is beyond the scope of the paper and requires one to use the total Hamiltonian and not just the energy density. This is an interesting area for future investigations and is mentioned in the text.

Comment

If the authors believe my claims about typical vs. ground states are incorrect, I will need to see a more careful analysis of the full spectrum of the Hamiltonian for some small system sizes. In fact, this may be a good idea to explore anyway in a new supplemental appendix. Is the spectrum extremely degenerate near $H=0$? Does random matrix statistics hold for the middle 20 percent of all energy levels? The method proposed in cond-mat/0610854 is quite numerically efficient and probably would work well to diagnose this question. If GUE/GOE etc. statistics are observed, this suggests that much of the spectrum is chaotic and not simple/single-particle.

Response

Our proof does apply to the full spectrum of H and not just the ground states. We find, however, exact diagonalization (ED) of $H(G_{ER}(p))$ and $H(G_{Complete})$ and comparison of their spectral statistics to be too limited by finite-size effects to be able to draw strong conclusions. The finite-size fluctuations are of order $(1/\sqrt{L})$ and as ED is limited to ~ 18 spins we are too far from the thermodynamic limit to perform a finite-scaling analysis. Nonetheless we have exploited the power of Matrix Product States in the new Section 3 of the SM to look at the scaling and convergence of $\langle H(G_{ER}(p)) \rangle$ for high energy states and system sizes up to $L = 200$. We observe the equivalence of this quantity to its complete-graph counterpart upon averaging over the Erdős–Rényi ensemble. We also witness how the variance of $\langle H(G_{ER}(p)) \rangle$ can increase with L before tending towards 0. This intriguing feature is mathematically hinted at during application of the Chernoff and Union bounds in our proof. We have made these parts of the proof more explicit in order to highlight this and collectively these results reinforce our statement (which is now included in the SM) that we need to access system sizes far beyond the limits of ED to reinforce our mathematical proof.

Referee 2

Main Comment

The paper is interesting, and I can imagine that the class of graphs identified in this work may attract attention of other researchers. While these results deserve to be published, I am uncertain at this stage concerning a recommendation for publication in Nature Communications. The main claim of the authors is essentially novel and important features of the models that they have identified, and I am not fully convinced. The main argument, as I understood it, is that in Fig. 5e and 5f the Shannon entropy almost reaches its maximal value at Δ_c . The authors argue that this is essentially different from the behavior for more conventional dense graphs, as shown in Fig. S3 of the Supplementary Material. At the same time, if one looks at the ER graph (left panels) in Fig. S3, the behavior is rather similar: nearly maximal value of the Shannon entropy close to Δ_c , with the difference that Δ_c scales with L (as a \sqrt{L}). This behavior of Δ_c for ER graphs is easy to understand in the considered model (the mean field solution favors ferromagnetism rather than antiferromagnetism for a finite Δ at large L); it can be different for other phase transitions. So, I am not persuaded by the degree of innovation and importance with respect to models on the graphs introduced in the paper. Can the authors make this more convincing?

Response

The Erdős–Rényi (ER) graph does maximize this Shannon entropy at the critical point. However, because this critical point vanishes towards $\Delta_c \rightarrow \infty$ in the thermodynamic limit we do not consider this a physical regime (it requires an infinite coupling strength) and thus the physics of the ER graph is relatively trivial compared to that of the irregular graphs, where the critical point does not vanish and there is a physically valid phase transition.

Importantly, what our results tell us is that for equilibrium states derived from $H(G_{ER}(p))$ (with finite microscopic parameters) the thermodynamic properties must be equivalent to the complete graph. Thus, the permutation-symmetry breaking statistical fluctuations present in the system cannot have any impact on the Physics and the system behaves as if it were completely homogeneous, with the diminishing Shannon Entropy as $L \rightarrow \infty$ serving as evidence of this. These results are not unique to the XXZ setup we considered for our computations but completely general for $H(G_{ER}(p))$.

We have added a significant amount of discussion, results and elaboration to the paper to make these points much clearer and stronger. For instance, in the SM, we have included calculations for the ground-state of the Transverse Field Ising model. We see that in the vicinity of the second-order critical point the ground state on the maximally irregular graph has non vanishing Shannon Entropy for the two point correlation matrix along both the x and z spin axes. We have compared these to finite-size and exact $L = \infty$ (which follow from our proof and the solution on the complete graph) results for the ER graph. Our finite-size ER results demonstrate convergence to these simple, exact results and show a diminishing Shannon Entropy across the phase diagram for increasing L . We have provided discussion around this in both the SM and the main text.

Additional Comments

1) In Fig. 4b and 4c, there are fits by power laws with some exponents (4.6 and 3.45). Do the authors propose that these fits should be taken seriously in the sense of asymptotic behavior?

Are there analytical arguments in favor of such a behavior? What is then expected for the exponents? Or is this just a numerical observation?

2) *Similar questions with respect to Fig. 5a and 5b.*

3) *Minor comment (I suppose, just typos):*

In Fig. 4b, in the fit, the constant +0.2 should be apparently -0.2 (minus sign)

And, similarly, in the fit in Fig. 5a, +0.35 should be apparently -0.35

4) *Fig. 5 caption:*

“... where each pair of vertices, other than two, have a different degree.”

Should this be “where each vertex ... has a different degree” ?

5) *In Supplementary Information, concerning the convergence of the DMRG calculation with MPS:*

“The truncation error is never greater than 1.2×10^{-4} . Combining this with the smoothness of the observables as a function of Δ suggests our results are reasonably accurate and well-converged.”

To demonstrate convergence more convincingly, it would be natural if the authors provide results for several values of the bond dimension (the maximal one used for the calculation and a couple of values smaller by a factor ~ 2).

Responses

1 + 2) The fits in these plots are numerical and we do not have any analytical arguments for their origin. Nonetheless such power-law scaling is observed for all system sizes we see and for two separate graphs - thus the qualitative behavior of these fits should be taken seriously. The explicit coefficients change significantly with system size and our Matrix Product State calculations are too limited in order to extract accurate values for the coefficients in the thermodynamic limit. We have made this explicit in the text.

3) This was an error, we have now put the appropriate negative signs on the constants here.

4) We have corrected this typo and thank the referee for spotting it.

5) We have, in the Supplemental, provided plots of the convergence of the ground-state energy density for several different values of the bond-dimension, up to the final value used. The convergence of these plots provides further reinforcement of the accuracy our results.

Changes to the text

Main Text

- 1) We have added a statement to the introduction emphasizing that our results apply in thermal equilibrium.
- 2) We have added an additional reference: *K. Xu et al, Probing Dynamical Phase Transitions with a Superconducting Quantum Simulator, Science Advances* **6** 25 (2020) to the introduction with demonstrates the realization of a dense graph (the complete graph) spin model on a quantum simulator.
- 3) We have clarified the physical meaning of the Hamiltonian in Eq. (1) as the total energy per spin of the system.
- 4) We have added a note emphasizing that our Hamiltonian has isotropic coupling strengths and so our conclusions do not extend to models such as the SYK model.

- 5) We have added a line discussing how our proof applies to the full spectrum of H , emphasizing equivalence of the free energy density for $H(G_{ER}(p))$ and $H(G_{Complete})$ at any temperature – not just for the ground state.
- 6) We have added a line referencing our new results in the SM which evidence our proof for much higher energy states and go beyond just ground-state calculations.
- 7) We have discussed our results on the Transverse Field Ising Model, referencing the non-zero, non-diminishing entropy we observe on the irregular graphs and how this is a feature not exclusive to the XXZ limit of our Hamiltonian.
- 8) We have removed several references (Refs. **19**, **22** and **38** in the original text) in order to make room for the, more pertinent, references added in this revision.
- 9) We have made the abstract more concise to reduce the word count.

Supplemental

- 1) We have improved the clarity of the proof of section 1. We have: **i)** provided a high-level description of the main steps of the proof, **ii)** made it explicit why we can treat all of the spin directions separately and that our proof applies to the full eigenspectrum of the operators, not just specific $SU(2)$ subspaces, **iii)** given an explicit example of the application of the Chernoff and Union bounds to bound the cut size N_{AB} and, finally, **iv)** made grammatical changes to improve the overall clarity of the section.
- 2) In Section 2 we have clarified that typical states in the spectrum of $H_{XXZ}(G_{Complete})$ have ~ 0 energy. We have added an equation for the degeneracy of the eigenstates of $H_{XXZ}(G_{Complete})$ and used this to provide a plot showing the probability density vs energy density for the spectrum of $H_{XXZ}(G_{Complete})$.
- 3) We have introduced a new section (Section 3) and corresponding plot (Fig. S1) in which we provide numerical evidence for the convergence of $H(G_{ER}(p))$ and $H(G_{Complete})$ when taking expectation values with respect to these higher energy typical states. We comment on how the non-monotonic behavior in the variance can be linked to results in our proof.
- 4) We have introduced a new section (Section 4) which discusses our choice of normalization for the Hamiltonian and its physical meaning in terms of the energy per spin. We discuss how our proof implies equivalence (between ER and complete) of any thermodynamic properties in equilibrium. We also discuss how non-equilibrium results require us to make statements on the total energy and thus are beyond the scope of the paper.
- 5) We have supplemented Fig. S4 with plots and discussion of the energy convergence versus sweep number in DMRG for values of Δ near the phase transition. This provides evidence that we still have an accurate measure of the GS properties even for the most difficult calculations we perform.
- 6) We have added a new section (Section 8) and corresponding plot (Fig. S6) where we provide results for the ground state energy of the Transverse Field Ising model on the maximally irregular and ER graphs. We have added several paragraphs discussing these results and how they reinforce our conclusions in the main text.
- 7) We have added two references: [6] *V. Oganesyan and D. Huse, Localization of interacting fermions at high temperature, Phys. Rev. B 75 15511 (2007)* and *J. Strečka*

and M. Jacscur and [10] A Brief Account of the Ising and Ising-like models: mean-field effective field and exact results, arXiv.1511.03031 (2015)

Throughout/ Other

- 1) We have added an affiliation to the lead author and edited the acknowledgements as their institution has now changed. We have expanded the addresses in the affiliation.
- 2) We have made sure all operators are notated with a hat symbol to avoid any confusion with scalars.
- 3) Grammatical and typographic changes which have improved the clarity of the manuscript but change none of our conclusions.
- 4) We have modified all plot labels with 'Energy' to 'Energy Density' in order to make it more explicit that our Hamiltonian's physical meaning is the energy per spin.

REVIEWER COMMENTS

Reviewer #1 (Remarks to the Author):

I thought that the revised manuscript was much improved. However, I am still a bit concerned about some of the remarks in the supplement that, as *operators*, $H(G) \rightarrow H(\text{complete graph})$ in the thermodynamic limit. This is a delicate statement because the Hilbert space is changing as part of the limit.

It seems like what the authors can argue for is that the operator norm of $H(G) - H(\text{complete})$ scales as $L^{-1/2}$, vs. the operator norm of $H(G)$ scaling as 1. That seems sensible to me. But then the authors state in the later supplement that what they can show is that the expectation value of H in any state is the same for $H(G)$ and $H(\text{complete})$. This is roughly the statement that the expectation value of $H(G) - H(\text{complete})$ is zero, which could be a trivial statement when the Hamiltonians are correctly scaled. But it's also true that almost every eigenvalue of $H(G)$ is 0 in the thermodynamic limit, meaning that for a typical state, the discrepancy between $H(G)$ and $H(\text{complete})$ might be important.

At a minimum I would suggest that the authors further add remarks along the lines I have suggested in this report, though I still think that some kind of numerical investigation. For example, is it not possible to compare $L=18$ and $L=10$ graphs and check whether or not a randomly chosen eigenstate from the middle of the spectrum has more or less overlap with a state of fixed S^2 ?

One can argue that the answer to these questions is a little distracting from the main punchline of the manuscript, which I think is mostly about the low-lying states in the spectrum, and is interesting! However the remarks I am making are important insofar as the authors want to say they prove the equivalence of the *entire spectrum* of the model, which includes those states where $H \rightarrow 0$ in the thermodynamic limit. So if the authors cannot convincingly demonstrate that there is no orthogonality catastrophe, they should soften their claims to state that the choice of graph G does not modify the spectrum only for the exponentially rare fraction of low-lying states in the spectrum.

Reviewer #2 (Remarks to the Author):

The authors have provided reasonable responses to most of questions / concerns in my first report and of the report by First Reviewers. They have also added the corresponding clarifications to the manuscript. I recommend the revised version of the paper for publication in Nature Communications.

Dear Editor and Referees,

We are pleased to hear that both referees feel the paper is much improved following the previous revisions and worthy of publication in Nature Communications.

In the below we respond to Referee 1's further concerns about our proof and its validity. We then provide a list of changes made to the manuscript in light of these concerns. We are very grateful for their critical reading of the proof as we believe this has helped us to clarify it significantly and be more mathematically precise.

Response to Referee 1

We agree with the referee that making statements about operators in the thermodynamic is a very delicate thing and not necessarily well-defined (in general properties of finite matrices do not immediately extend to infinite ones). As a result, we no longer talk about the whole spectrum of $H(G_{ER}(p))$ and $H(G_{Complete})$ for $L \rightarrow \infty$, but instead consider equivalence for thermodynamic properties in thermal equilibrium only.

Specifically, we have gone back through our proof, and working from within the finite size limit and a specific basis, rigorously shown that the free energy densities on the ER vs Complete Graph converge towards each other as one increases the system size. We achieve this by several modifications to our original proof:

1. We re-define $H(G)$ as the total energy of the system by scaling its original form by a factor of L .
2. We use our previous strict bound on cut-sizes of ER graphs to prove that the magnitude of the largest eigenvalue of the difference matrix $H(G_{ER}(p)) - H(G_{Complete})$ scales as $O(\sqrt{L})$. $H(G_{ER}(p))$ and $H(G_{Complete})$ meanwhile each have eigenvalues that can scale as large as $O(L)$.
3. This directly allows one to prove that the absolute difference in the free energy density $\left| f(H(G_{ER}(p))) - f(H(G_{Complete})) \right|$ scales as $O(L^{-\frac{1}{2}})$ and thus vanishes as $L \rightarrow \infty$. We define $f(H(G))$ as $f(H(G)) = -\frac{1}{\beta L} \ln(\text{Tr}(\exp(-\beta H(G))))$.

We believe that these results are much more mathematically precise and avoid making statements about the eigenspectrum of operators directly in the thermodynamic limit. We have completely re-worked Section 1 of the Supplemental Material to reflect these statements, introducing Theorems and Lemmas in order to be as clear as possible.

Furthermore, we have performed computational calculations which explicitly demonstrate point 2. above – i.e. that the magnitude of the largest eigenvalue of the difference matrix $H(G_{ER}(p)) - H(G_{Complete})$ scales as $O(\sqrt{L})$. We achieve this with ED calculations for $L \leq 16$ and DMRG calculations up to $L = 30$. These calculations are done for a range of parameters and not specific to any limit of $H(G_{ER}(p))$.

As we can prove the convergence of the free energy density, our results are not just about low energy states but do apply to thermodynamic equilibrium at any temperature. The referee makes a significant point about the $O(\sqrt{L})$ fluctuations, however. These are likely to not disappear when measuring local observables or during out-of-equilibrium dynamics. This is a fascinating avenue for future research and something we have made more explicit in the main text.

Changes Made to the Manuscript

Main Text

- 1) Eq. (1) for the total Hamiltonian has been re-scaled so that it's meaning is the total energy. The relevant text has changed too.
- 2) Eq. (2) in the main text has been modified to a theorem which states that the difference in free energy densities converge to within $O(1/\sqrt{L})$. The surrounding discussion has been modified to reflect this and we no longer make statements about the eigenspectrum of the Hamiltonians directly in the thermodynamic limit.
- 3) We have added to the acknowledgements our gratitude to the referees for helping substantially improve our manuscript.

Supplemental

- 1) Section 1.1 has been changed to follow and rigorously prove the statements **2.** and **3.** above – Lemmas and Theorems have been introduced to this effect.
- 2) Section 1.5 has also changed to state and prove a theorem about the free energy density on random graphs with non-trivial cuts.
- 3) What was Section 3 before (Numerical Verification of the Proof) has been moved to Section 2. The results we introduced in the last revision have now been replaced with direct verification of point **2.** above – we believe this is more meaningful than the previous numerical results.
- 4) What was Section 4 (Normalisation and Physical Meaning of $H(G)$) has now been removed and its content has been woven into the main text and Supplemental for ease of reading.

General

- 1) Further grammatical and wording changes to improve the readability of the manuscript.
- 2) We have replaced the notation $\beta_x, \beta_y, \beta_z$ with w_x, w_y, w_z to avoid any confusion with the inverse temperature β .

Dr. Joseph Tindall

REVIEWERS' COMMENTS

Reviewer #1 (Remarks to the Author):

The authors have made sufficient changes. I am happy with publication in present form.